# Effect of Rockfall Spatial Representation on the Accuracy and Reliability of Susceptibility Models (The Case of the Haouz Dorsale Calcaire, Morocco)

**Youssef El Miloudi** [1] , **Younes El Kharim** [1] , **Ali Bounab** [1] **and Rachid El Hamdouni** [2],*

[1] Laboratoire de Géologie de L'Environnement et Ressources Naturelles (GERN), Faculté des Sciences, Abdelmalek Essaadi University, Tetouan 93000, Morocco; youssef.elmiloudi@etu.uae.ac.ma (Y.E.M.); yelkharim@uae.ac.ma (Y.E.K.); ali.bounab@etu.uae.ac.ma (A.B.)

[2] Civil Engineering Department, E.T.S. Ingenieros de Caminos, Canales y Puertos, Campus Universitario de Fuentenueva, s/n Granada University, 18071 Granada, Spain

* Correspondence: rachidej@ugr.es

**Abstract:** Rockfalls can cause loss of life and material damage. In Northern Morocco, rockfalls and rock avalanche-deposits are frequent, especially in the Dorsale Calcaire morpho-structural unit, which is mostly formed by Jurassic limestone and dolostone formations. In this study, we focus exclusively on its northern segment, conventionally known as "the Haouz subunit". First, a rockfall inventory was conducted. Then, two datasets were prepared: one covering exclusively the source area and the other representing the entirety of the mass movements (source + propagation area). Two algorithms were then used to build rockfall susceptibility models (RSMs). The first one (Logistic Regression: LR) yielded the most unreliable results, where the RSM derived from the source area dataset significantly outperformed the one based on the entirety of the rockfall affected area, despite the lack of significant visual differences between both models. However, the RSMs produced using Artificial Neural Networks (ANNs) were more or less similar in terms of accuracy, despite the source area model being more conservative. This result is unexpected given the fact that previous studies proved the robustness of the LR algorithm and the sensitivity of ANN models. However, we believe that the non-linear correlation between the spatial distribution of the rockfall propagation area and that of the conditioning factors used to compute the models explains why modeling rockfalls in particular differs from other types of landslides.

**Keywords:** rockfall; susceptibility; propagation area; logistic regression; artificial neural network



## 1. Introduction

Landslide risk and hazard assessment is an essential step in land use mapping, land management efforts, and urban planning, especially in mountainous regions. Of the various landslide types, rockfalls and rock avalanches have the highest death rates due to their high velocity, high kinetic energy, and long runout distances [1,2]. According to [3,4], such landslide processes mainly involve large boulders and thus occur almost exclusively in rocky domains. In Morocco, 52 people were killed in 1988 as a result of a cliff collapse in Fes City [5], an event similar to the one that occurred in Cairo, Egypt in 2008. The Rif Mountain chain is the most landslide-prone region in Morocco, particularly in the central Rif [6–8]. In this environment, rock falls occur mainly along the coastal cliffs of the Mediterranean coast [9,10] and along the Dorsale Calcaire escarpments [11]. In the study area (the northern segment of the Dorsale Calcaire morpho-structural unit), large boulders buried a house in the village of Onsar (35°38′54″ N; 5°25′47″ W), leading to the death of its occupants in the early 20th century [12]. Similar rockfalls reoccurred in 1963 and later in the year 2000, which pushed the authorities to temporarily evacuate the village shortly after the events. Another rockfall-prone sector in the study area is the Bouanane cliff, which has

produced recurring rockfall events since the early 1990s [13]. As of 2024, there have been no reported casualties in Bouanane. Nevertheless, the size of the boulders, which may exceed 5 m (1994 event), is cause for concern, especially given the damage endured by two passenger vehicles in 2011 following the detachment of three mid-size boulders (1 to 2 m in diameter) [13]. Apart from the inherent rockfall incidents in the study location, the rockfall threat is further intensified by human activities in other regions of the study area. Changes were made to the natural topography of the terrain by old and new quarries, which compromise the stability of relicts and mature processes [14].

In this respect, it is necessary to evaluate rockfall spatial representation's effect on susceptibility models' accuracy and reliability. As such, two datasets are used to build two susceptibility models: the first includes only the source area (S) and the second involves the entirety of the rockfall (i.e., source + deposition area (S + PA)). The study area corresponds to the Dorsale Calcaire morpho-structural unit, which outcrops in Northern Rif regions [15–18]. This geological domain consists mainly of massive limestone and dolomite layers outcropping as N–S-oriented, mid- to high-altitude cliffs. The latter are formed by the plio-quaternary erosion of West-verging thrust sheets during interglacial periods [19,20]. Therefore, an abundance of rockfall processes can be inventoried, which constitutes a statistically significant sample for our study. In addition, the above-described socio-economic impact of such frequent rockfall occurrences requires a good understanding of their spatial distributions, which we also attempt in this paper.

## 2. Study Area

The study area is large, covering over 350 km². It is part of the Dorsale Calcaire morpho-structural unit, which dominates over the western and central Rif chain relief [21–23]. From a structural standpoint, it is located in an intermediate position between the Paleozoic outcrops to its east (Ghomarides units) and the External Rif domain to its west. In previous research [18–25], the Dorsale Calcaire is conventionally split into three main blocks. The first one is located between the Gibraltar Straight (northern boundary) and the Tetouan major tectonic accident (southern boundary). The second corresponds to the Dorsale Calcaire thrust sheets that outcrop between Tetouan and Assifane [18]. Finally, the third block can be located further to the East (Dorsale Calcaire of Nekkor) in the Al Hoceima region.

This research mainly focuses on the first block, also known as the Haouz Dorsale Calcaire subunit (Figure 1). In addition to the major thrust faults that dominate the Dorsale Calcaire unit, the Haouz sub-unit is exclusively characterized by back-thrust and locked thrust structures, especially near the Gibraltar strait [15–26]. In geomorphological terms, the Haouz subunit is marked by lower elevation, base level, and mean slope values compared to the central and southern segments of the Dorsale Calcaire [19–21]. However, its N–S-oriented limestone ridges dominate over the entirety of the internal domain and are known for their abundance of rocky cliffs and rockfall processes (Figure 2) [16].

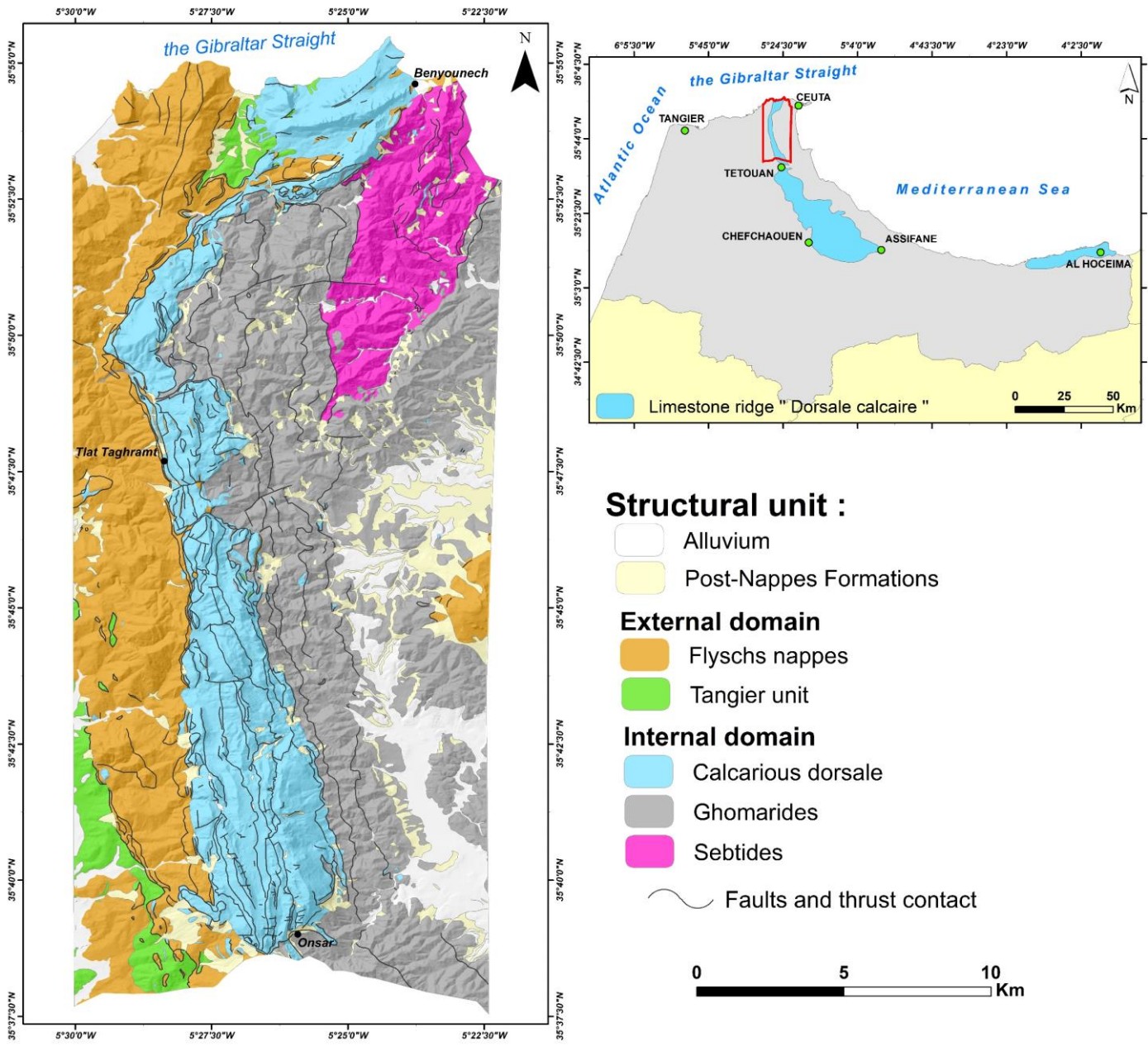

**Figure 1.** Geological map of the study area.

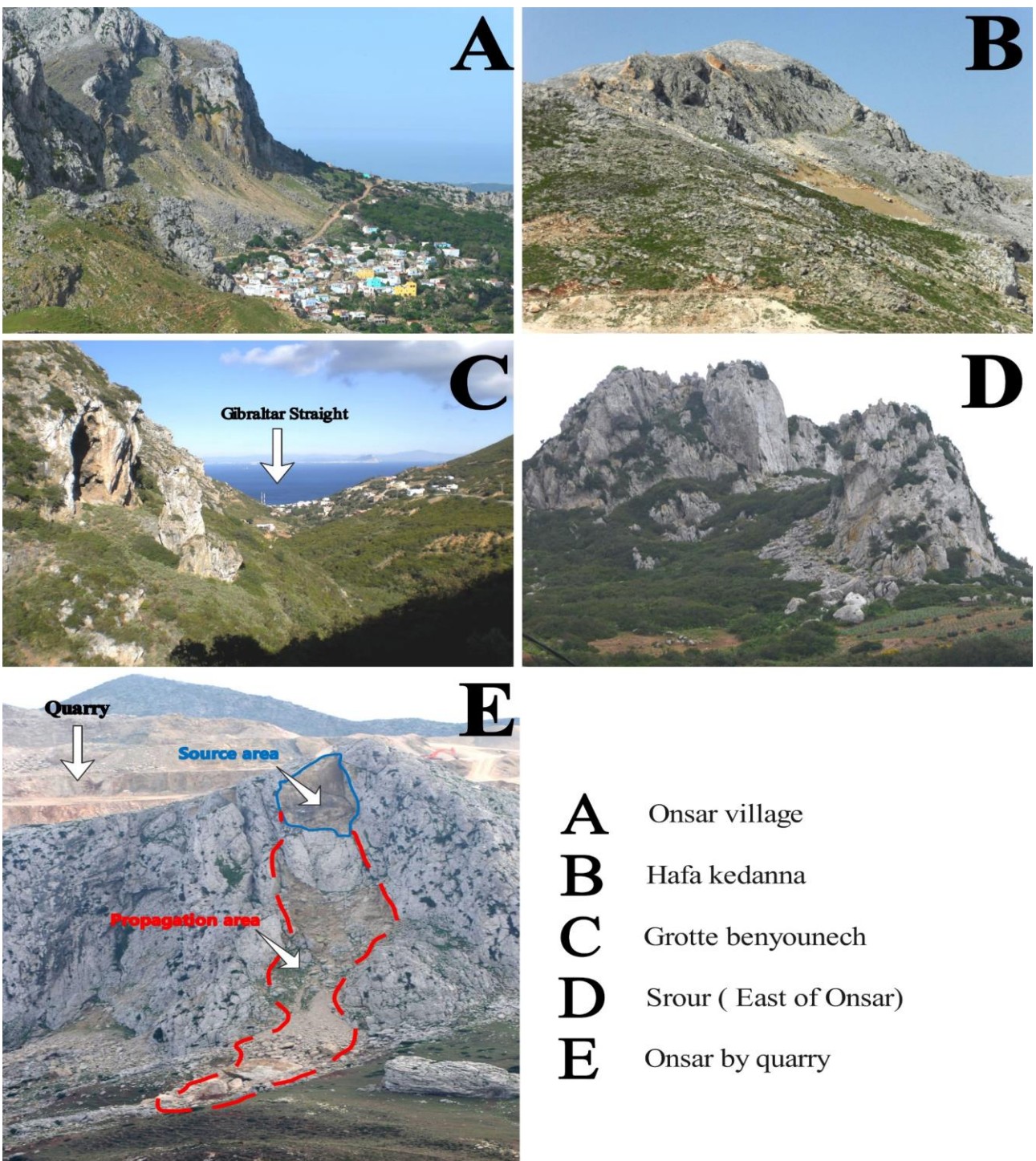

**Figure 2.** Examples of rockfall processes in the study area.

## 3. Materials and Methods

As stated before, this work attempts to build rockfall susceptibility models (RSMs) covering the study area using two explanatory statistics techniques. The first technique is logistic regression (LR), which is a widely used multivariate statistical model that yields the most reliable and consistent results with the least sensitivity to input data variations [27–30]. The second is the multilayer perception approach (MLP), which requires no prior knowledge of the investigated phenomena or preprocessing of the explanatory variables [31,32].

The training data (i.e., 70% of the rockfall inventory) are split into two datasets depending on the rockfall representation approach used for the inventory. The first set is

composed of polygons that exclusively cover the source area of each individual rockfall, while the second dataset covers the entirety of the rockfall processes identified within the borders of the study area (Figure 3). This strategy allows us to assess the effect of rockfall representation on the reliability and consistency of the results. Furthermore, the use of two techniques should reveal any method-specific biases that can either exaggerate or minimize such effects. With regard to RSM validation, the remaining 30% of the inventoried rockfalls are used to build the testing dataset (Figure 3).

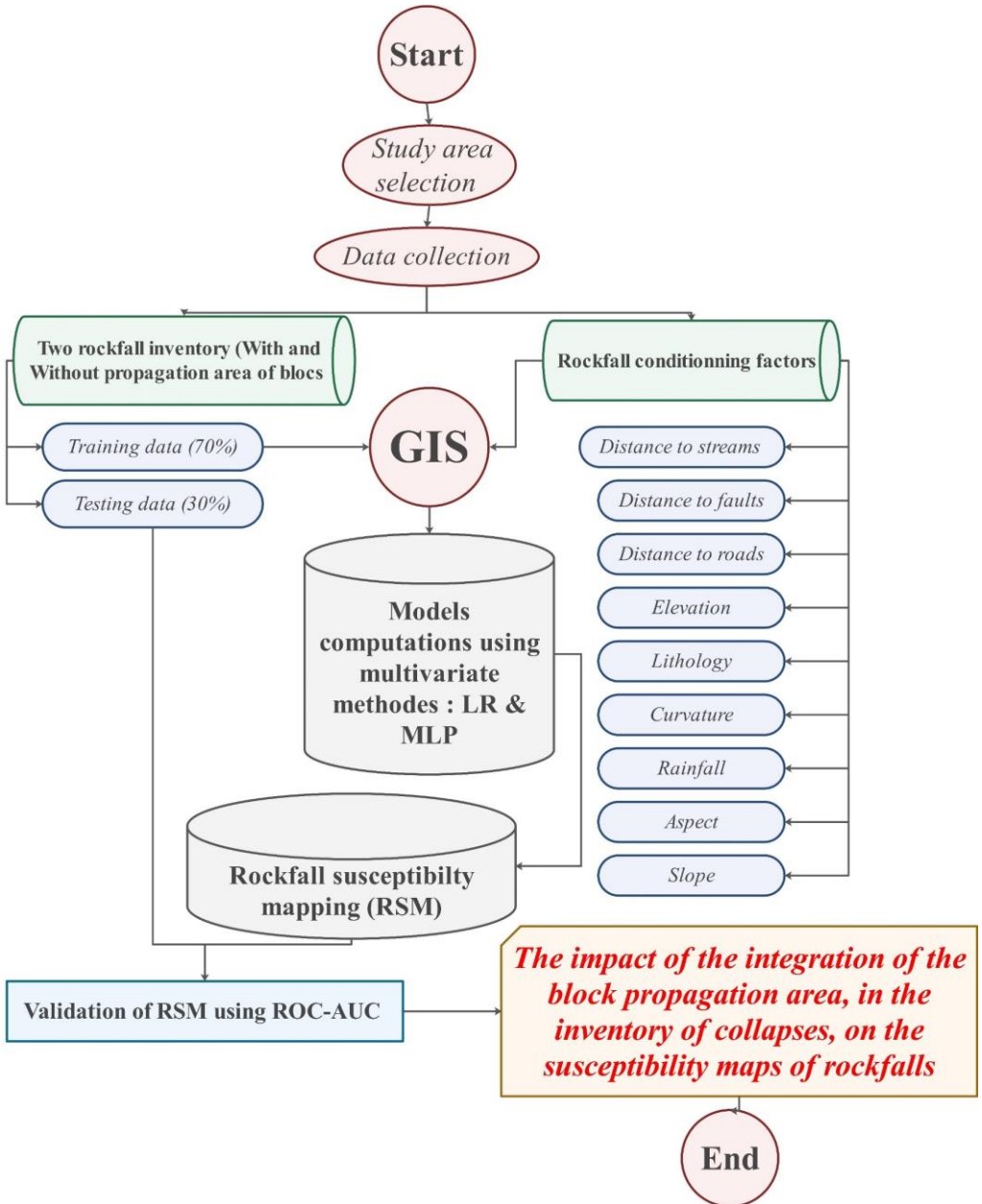

**Figure 3.** Methodology flowchart followed in this study.

### 3.1. Logistic Regression

Multiple logistic regression evaluates the likelihood of a rockfall event using several physical criteria [33]. It establishes a connection between the existence or absence of an investigated phenomenon and its explanatory variables using the maximum likelihood approach [34]. The dependent variable used in this work is binary, expressing the pres-

ence/absence of previous rockfalls. The expression of the logistic regression model is presented in Equation (1).

$$log(y) = \beta 0 + \beta 1 X1 + \beta 2 X2 + \cdots + \beta i Xi + E \tag{1}$$

Thus, the probability of occurrence of a rockfall is calculated using Equation (2):

$$RSI = (P) = \frac{1}{1 + e^{-(log(y))}} = \frac{1}{1 + e^{-(\beta 0 + \beta 1 X1 + \beta 2 X2 + \cdots + \beta i Xi + E)}} \tag{2}$$

where *RSI* is the rockfall susceptibility index, *P* is the probability of rockfall occurrence, *y* is the dependent variable, $\beta 0$ is a constant, $\beta i$ is the *i*th regression coefficient, *Xi* is the explanatory variable, and *E* is the error.

However, using different measurement scales and units can compromise the reliability of the regression coefficients (Equation (2)) and odds ratio (OR) estimation (Equation (3)). Therefore, a z-score standardization was performed on all independent variables. Subsequently, OR values were computed using Equation (3) to assess the association between the spatial distribution of each explanatory variable/factor and that of rockfall occurrence.

$$OR = \frac{p}{q} \tag{3}$$

where *p* and *q* are the presence/absence probabilities of rockfalls, respectively, as determined by the regression model, and *OR* is the odds ratio coefficient.

### 3.2. ANNs Built Using the Multilayer Perception Technique (MLP)

Artificial Neural Network analyses are based on powerful algorithms that emulate the functioning of human brain cells. This technique is part of the soft computing techniques that use evolutionary genetic algorithms, which differ from traditional methods in their ability to analyze extremely complex phenomena [35,36].

Such highly adaptive and flexible algorithms [37] consist of multiple layers of interconnected neurons that use the provided training dataset to solve a specific problem. To achieve this, the algorithm modifies the connection weights to reduce the error between the expected and projected output [38]. Multi-layer neural networks, or MLPs, are considered the ideal choice for landslide studies due to their standard error of around 0.14, which is lower than that of other network structures [39]. The three layers of the MLP processing chain are the input layer, the output layer, and one or more hidden layers. The performance of the model improves through a process of trial and error until the optimal outcome is obtained [29].

Thus, the *RSI* value for each cell is presented in Equations (4) and (5).

$$Net = \sum_{i=1}^{n} xi \, wi \tag{4}$$

$$RSI = Pj = \frac{1}{1 + e^{-Net}} \tag{5}$$

Each cell in the study area has a corresponding pseudo-probability value denoted by *Pj*, and each neuron in the hidden layers of the model receives an input represented by Net.

### 3.3. RSM Accuracy Assessment

Based on the area under the receiver operating characteristic (ROC) curve (AUC), the output RSMs' performance is evaluated and is obtained by plotting sensitivity values as a function 1-specificity [40]. Sensitivity and specificity measurements are crucial tools for evaluating the accuracy of probabilistic models. Equation (6) illustrates how to compute sensitivity, which is the number of true positives divided by the sum of true positives and false negatives. On the other hand, specificity is determined by dividing the total number of

true negatives by the sum of true negatives and false positives. It quantifies the percentage of correctly recognized genuine negatives (as seen in Equation (7)).

$$Sensitivity = \frac{TP}{TP + FN} \qquad (6)$$

$$Specificity = \frac{TN}{FP + TN} \qquad (7)$$

where *FP* stands for false positives, *FN* for false negatives, *TP* for true positives, and *TN* for true negatives.

After building the ROC curve, the AUC value is estimated for each model, which ranges from 0 to 1. Accordingly, a model can be considered either bad (AUC < 0.5), average (0.5 to 0.6), moderate (0.6 to 0.7), good (0.7–0.8), very good (0.8–0.9), or excellent (0.9–1) [29–41].

## 4. Database of Rockfall Inventory and Conditioning Factors

### 4.1. Rockfall Inventory

The data used in this study are derived from various sources. For the rockfall inventory effort, a set of aerial photographs dating back to 1966 and 2010 were used to identify old processes. In addition, multidate, satellite-derived orthoimages provided by Google Earth were used to recognize newer rockfalls and those related to recent manmade modifications to the natural topography. The resulting inventory included a total of 125 rockfalls, most of which were visited during field surveys. As stated before, a Geographic Information System (GIS) was used to produce a polygon shapefile that exclusively maps the source area of the identified rockfalls (Figure 4A) and another one that covers the entirety of the inventoried processes (Figure 4B).

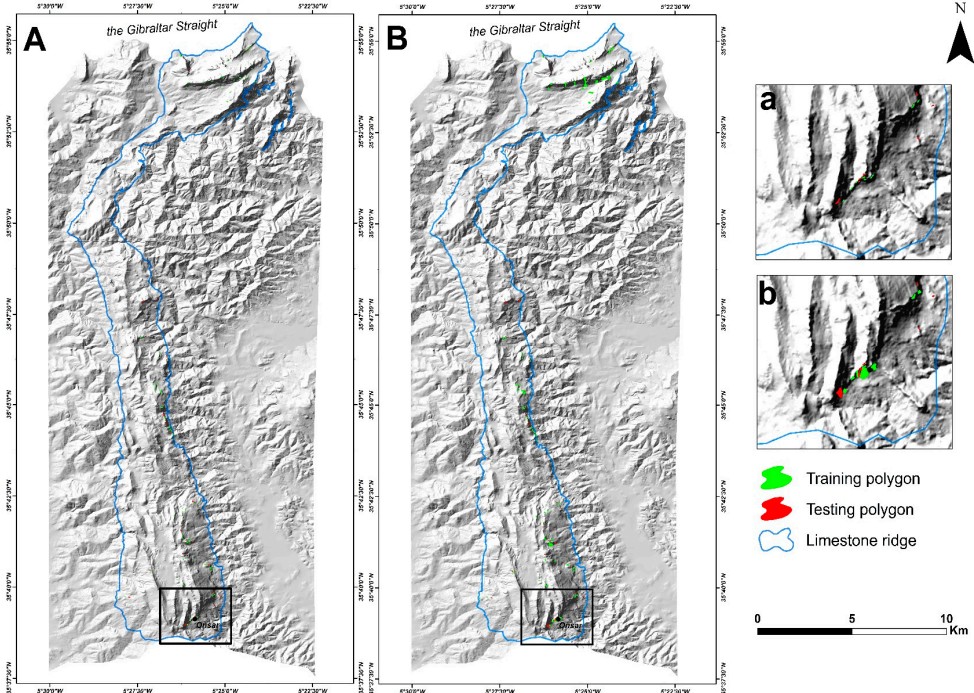

**Figure 4.** **A** (**a**)—Rockfall inventory map showing exclusively the source area. **B** (**b**)—Rockfall inventory map showing the entirety of the inventoried rockfalls.

In total, an area of 0.029 km² corresponds to rockfall sources, which is significantly smaller than the area covered by the full inventory (0.243 km²). However, only 0.021 km² of the first inventory and 0.2 km² of the second were used to build the model (i.e., 70% of the

dataset). These data are thus deemed imbalanced concerning the proportion of rockfall vs. non-rockfall pixels/area. In fact, less than 0.07% of the total investigated area is involved, despite the Haouz subunit hosting a relatively large number of rockfall occurrences.

### 4.2. Rockfall Conditioning Factors

#### 4.2.1. Slope

Rockfalls occur on near-vertical rocky slopes [42]. To accurately map such sub-vertical surfaces, a slope raster layer (Figure 5A) was obtained from a 5 m resolution digital elevation model (DEM) using a GIS plate form. The DEM itself was obtained by interpolating the digitized contour lines of the official 1/25,000 topographic maps of the study area. This high resolution relative to the size of the study region provides more detailed information than the 30 m-SRTM models often used in RSM research.

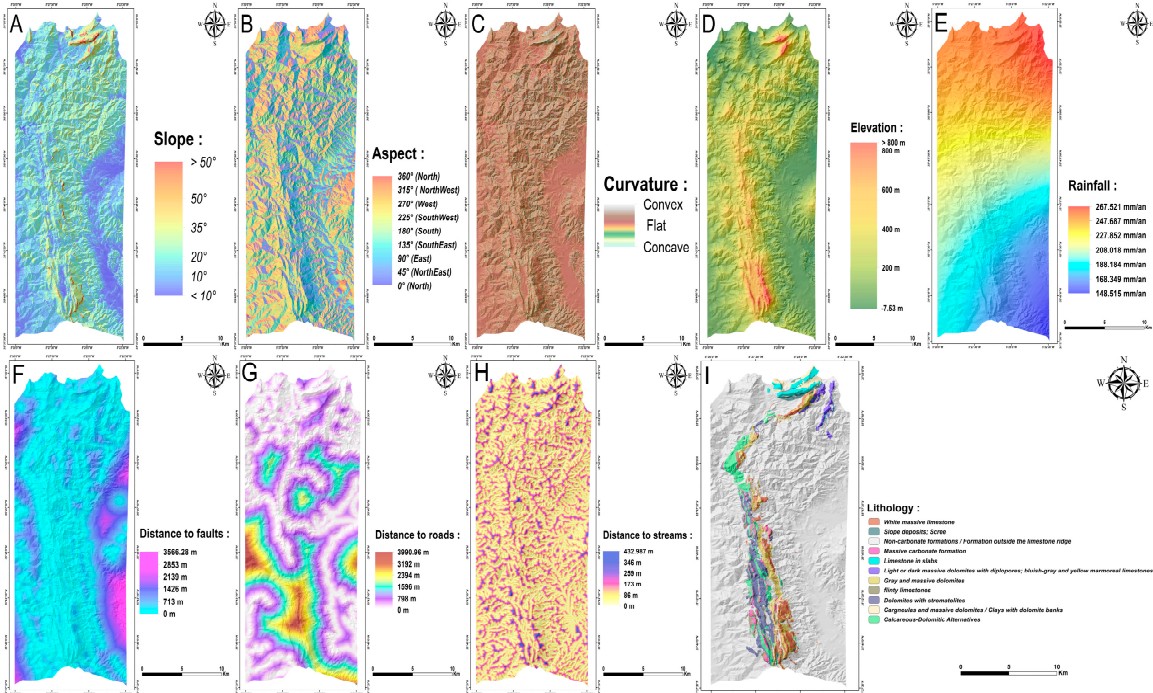

**Figure 5.** Rockfall conditioning factors ((**A**): slope; (**B**): aspect; (**C**): curvature; (**D**): elevation; (**E**): rainfall; (**F**): distance to faults; (**G**): distance to roads; (**H**): distance to streams and (**I**): lithology).

#### 4.2.2. Aspect

In addition to slope steepness, the orientation of a given slope also influences its stability. In fact, solar radiation in the temperate climate zone is highly dependent on the season and the slope's aspect. The latter variable controls the amount of radiation received by the north-facing and south-facing slopes, where the former type receives less radiation than the latter [43]. This leads to a significant difference in groundwater content between both slope types. Therefore, the slope aspect (Figure 5B) must be included as a potential predictive variable. In this case, study, it is derived from the same 5 m DEM described in Section 4.2.1.

#### 4.2.3. Curvature

In addition to the above variables, a slope's geometry also plays a crucial role in determining surface and groundwater flow. As such, two categories can be distinguished (Figure 5C): concave slopes that tend to slow surface runoff and promote infiltration, and convex surfaces that generate the opposite effects. To quantitively assess this geometry the curvature variable is conventionally used [44], which can be automatically computed

using GIS tools. In this case, study, the same 5 m DEM is used as input for the curvature computation algorithm (See Section 4.2.1).

### 4.2.4. Elevation

Elevation may promote the occurrence of landslides in some geomorphological and climatological conditions [45]. This is because precipitation and/or windspeed are dependent on elevation. As such, the abovementioned 5 m-DEM is an elevation raster layer covering the entirety of the study area, which can be used as is to represent the spatial distribution of this variable (Figure 5D). Similar to the slope raster layer, elevation is sensitive to microtopographic variability, which can be captured using a higher-resolution raster.

### 4.2.5. Rainfall

Rainfall patterns are an undeniable factor in the occurrence of land movement. Rainwater infiltration causes pore pressure to rise and soil suction to decrease on slopes; the weight will also increase, meaning soil shear will reduce. The results provided by [35] show that the frequency of minor rock instabilities increases significantly due to winter freeze–thaw cycles and moderate precipitation. When assessing landslide susceptibility, only the spatial component of precipitation is taken into account [46,47]. The precipitation map (Figure 5E) used in this analysis is produced using a linear equation with the elevation of the study area and the annual precipitation data between 2010 and 2021 from NASA after IDW interpolation.

### 4.2.6. Distance to Faults

The presence of major faulting promotes the formation of complex tectonic landforms. In the Haouz subunit, major sub-horizontal thrust faults seem to control the geological structure and also promote the formation of sub-vertical cliffs alongside the said faults. This is because differential erosion processes exploit the lithological heterogeneity alongside such fault lines, which promotes slope instability processes. Therefore, a distance to the major faults' raster layer (Figure 5F) was produced using a shapefile of the thrust and strike–slip faults mapped and documented in the 1/50,000 geological maps of the study area.

### 4.2.7. Distance to Roads

Given the recent mining activity in the Haouz Dorsale Calcaire, some roads were built to connect the newly opened quarries to the major roads of the area, which introduced significant modifications to the topography. Therefore, the distance to the road (Figure 5G) was used as a predictive variable for our RSM computation effort. The road network used to build the input raster layer is provided by the Ministry of Equipment and Logistics of Morocco (METL).

### 4.2.8. Distance to Streams

Water stream erosion steepens hillslopes, which consequently promotes landslides. A study by [48] shows that the frequency of landslides generally decreases as the distance from water streams increases. In this regard, we used a water stream shapefile to generate a distance to the stream raster layer using a GIS platform (Figure 5H). Although stream erosion plays a less important role in rockfall dynamics, it may generate steep-to-sub-vertical slopes in the geomorphological setting of the Dorsale Calcaire morpho-structural unit. Therefore, it is included in the analysis as a potential conditioning factor.

### 4.2.9. Lithology

Lithology is widely used in landslide susceptibility research. It is considered to be the most influential conditioning factor by many authors working in various geological settings [30–49]. Despite its apparent lithological homogeneity (mainly formed by massive carbonate rocks), the Dorsale Calcaire unit consists of limestones and dolostones that are affected by various degrees of karstification, which consequently induces different slope

dynamics. Its lithological formations also vary in terms of thickness, with Hettangian and Sinemurian massive formations presenting the highest and steepest cliffs and consequently housing the most rockfall occurrences [12]. Therefore, a lithological map (Figure 5I) derived from digitizing the 1/50,000 geological maps is used as a conditioning factor to forecast the spatial probability of landslide occurrence (i.e., susceptibility).

## 5. Results

### 5.1. Rockfall Susceptibility Maps (RSM)

The logistic regression models derived from the source area and the full rockfall inventories are presented below (Equation (8)). In general, the source area model proves the existence of an inversely proportional correlation with the distance to fault variable and a strong proportionality with slope steepness. This indicates that steep slopes and major fault lines are statistically associated with rockfalls. Other variables present a low correlation with rockfall occurrence.

$$
\begin{aligned}
\log(\text{Source}) = {}&-14.206 + 0.717 \times \text{C.slope} - 0.193 \times \text{C.aspect} + \\
&0.058 \times \text{C.curvature} + \text{C.lithology} - 0.162 \times \text{C.rainfall} - 2.499 \times \\
&\text{C.distancetofault} + 0.042 \times \text{C.distancetoroad} + 0.034 \times \\
&\text{C.distancetostream} + 0.199 \times \text{C.elevation}
\end{aligned}
\tag{8}
$$

The model that includes the propagation and deposition area yields slightly similar results for the LR technique. The slope and distance to fault variables present the highest positive and negative correlation values, respectively. However, the correlation coefficient values (Equation (9)) are lower in the second model, which shows that including the propagation zone does not significantly affect the statistical tendencies of the model, but rather affects the quantitative assessment of the correlation coefficients.

$$
\begin{aligned}
\log(\text{Wholerockfall}) \\
= {}&-13.013 + 0.661 \times \text{C.slope} - 0.180 \times \text{C.aspect} \\
&- 0.038 \times \text{C.curvature} + \text{C.lithology} - 0.105 \times \text{C.rainfall} \\
&- 1.739 \times \text{C.distancetofault} + 0.172 \times \text{C.distancetoroad} \\
&- 0.070 \times \text{C.distancetostream} + 0.330 \times \text{C.elevation}
\end{aligned}
\tag{9}
$$

The output RSM maps show evidence for slight variability, where the first model (which exclusively considers the source area) produces more conservative results compared to the second one (Figure 6A,B). About the ANN models, the results show more variability when the deposition/propagation area is included (Figure 6C,D). The ANN model based on polygons covering the entirety of the rockfalls in the study area overestimates the RSI values. Consequently, the ANN algorithm is found to be more sensitive to input data variability in comparison with the LR models.

In statistical terms, the frequency distribution of very low, low, medium, high, and very high categories for both the LR RSMs (Figure 7A) suggests a significant difference between the first and second categories. These two categories cover most of the study area, which shows the rareness of rockfall occurrences in the Haouz subunit. Conversely, the medium, high, and very high susceptibility distributions reveal no significant variability in terms of frequency. However, they cover very small portions of the study area for both models. As for the ANN algorithm, the difference is more relevant in the high to very high categories, with the source + propagation training data producing more liberal models, with more abundant high susceptibility pixels (Figure 7B). However, the very low susceptibility class is still the most dominant, with a spatial coverage of more than 94% in both RSMs.

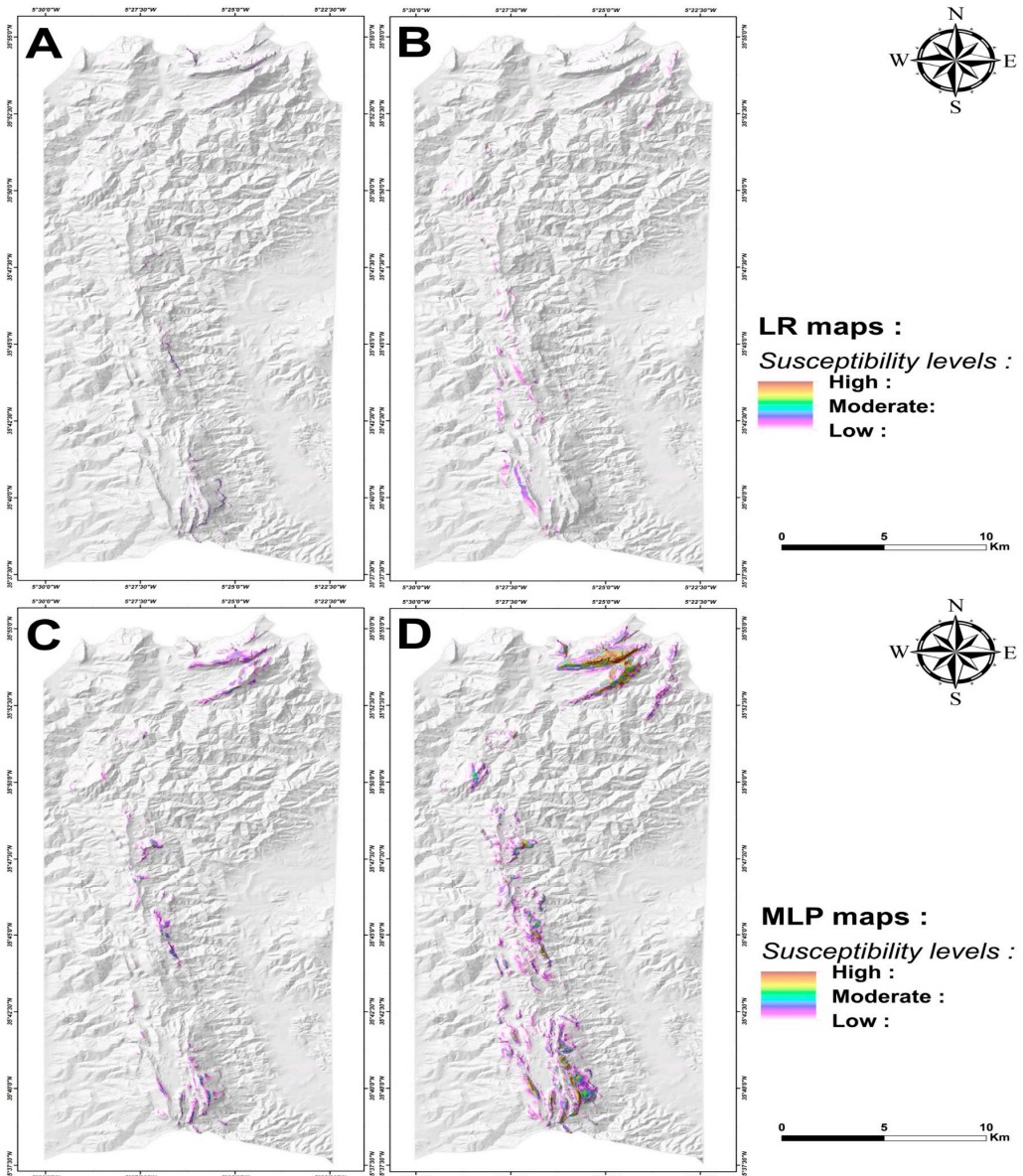

**Figure 6.** Rockfall susceptibility maps were produced using logistic regression and artificial neural networks. ((**A**,**C**): RSM based on source area only; (**B**,**D**): inventory based on the entirety of the rockfalls).

An analysis was carried out to explore the degree of influence of each variable (causal factors) on both MLP output maps. The results are shown in Figure 8. Our findings show that the most influential variables vary according to the inventory approach used to generate the input. The variables, namely slope, aspect, distances to faults, and lithology, had similar degrees of importance for both maps (Figure 6C,D), with the slope steepness variable being the most significant. On the other hand, the degree of importance of variables such as curvature, elevation, rainfall, distance to roads, and distance to streams changed significantly after adding the propagation and deposition zones to the inventory (Figure 8).

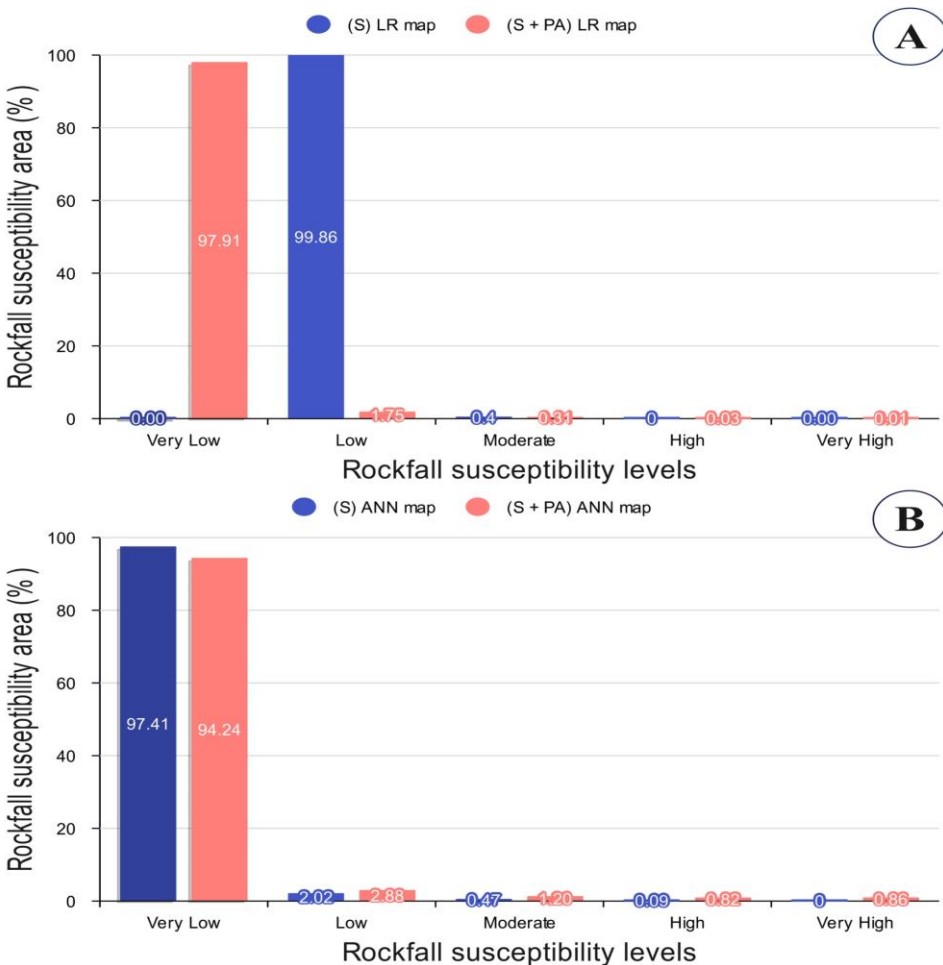

**Figure 7.** Histogram of rockfall susceptibility categories. Histogram (**A**) shows logistic regression (LR) and histogram (**B**) shows the artificial neural network (ANN) using MLP approach. (S) is the designed source, and (S + PA) is the designed source + propagation area.

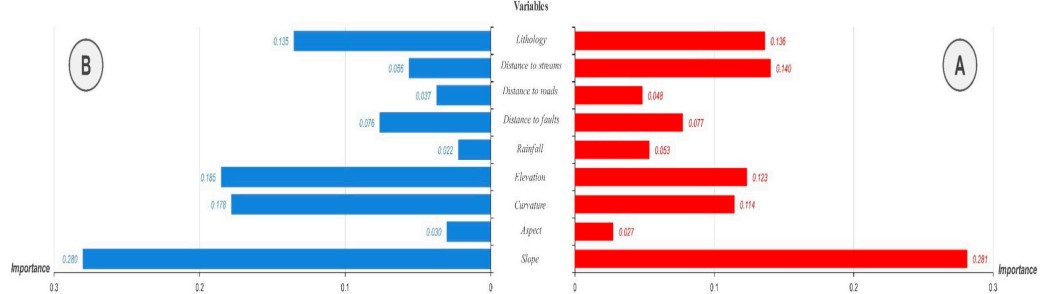

**Figure 8.** Importance degree of causative factors in MLP maps ((**A**) for the (S) ANN map and (**B**) for the (S + PA) ANN map).

### 5.2. RSM Accuracy Assessment

RSM validation is conducted using 30% of the dataset. The ROC curves shown in Figure 9 indicate that the AUC value differs when the rockfall-propagation zone is included in the input training data.

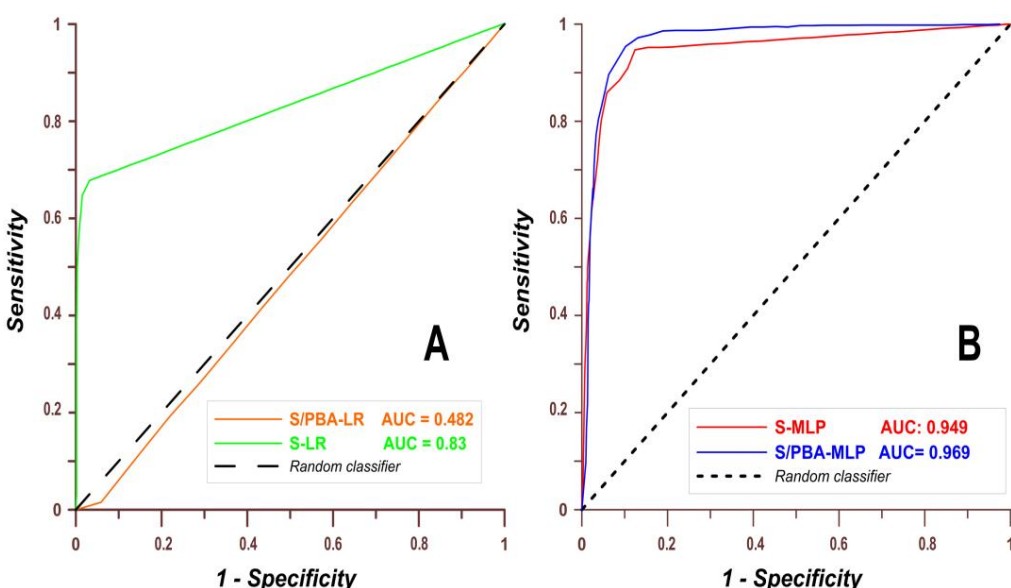

**Figure 9.** ROC curves for the output RSMs: (**A**)—logistic regression (LR) curves: (S/PBA); (**B**)—multilayer perception (MLP) curves, with (S) being the designed source and (S/PBA) being the source + propagation area.

The LR technique allowed for building a very good predictive model with an AUC value of 0.83 when the source area was exclusively used as the input. However, the RSM that uses the propagation zone as input can be considered poor, with an AUC of approximately 0.49 (Figure 9A). This large difference in AUC values is not reflected in the RSI geographic distribution maps, since the latter do not reveal as much difference between both databases. As for the ANN algorithm, both models can be considered excellent, given their high AUC values that exceed 0.9 (Figure 9B). The difference between both RSMs in terms of accuracy assessment is small despite the significant variability in the geographic distribution of the output susceptibility categories.

## 6. Discussion

Based on the results, including or deleting a portion of the input polygon surface can significantly impact the output model both in terms of geographic and frequency distributions as well as accuracy assessment. However, the amount of change and its significance can vary depending on the technique used to build the model. In previous research, LR was shown to produce the most reliable results both in terms of RSM category distribution and AUC values [29,30].

In this case, study, the AUC values assigned to the LR models resulted in different accuracy assessments despite the similar correlation coefficients and spatial distribution of RSI values. One LR (S) model can be classified as very good, while the other (S/PBA) is deemed to be poor. However, ANN produced consistent AUC values despite utilizing different inputs. This is in contrast to previous studies wherein LR algorithms were considered robust. To explain this anomaly in our findings, one can refer to the particularity of rockfalls, which are rarer than other types of landslides. In addition, mapping such processes is not a settled issue because of the large difference between the physical characteristics of the source and deposition areas.

In the MLP models, the number of hidden layers produced by the algorithm, which is a very important indicator of the degree of complexity of a given model [38], is the same for both training datasets. This means that including the propagation zone as input does not change significantly the complexity of the model to the point where hidden layer estimation is largely affected. However, the effect of some variables on the RSI estimation process was significantly changed because of the large contrasts between rockfall scarps and deposition areas in terms of geomorphological features. Although the geographical distribution and

frequency of RSI values suggest that the results are different, the AUC assessment results remained consistent unlike those of the LR algorithm.

These contradict previous research where the ANN is considered the most sensitive RSI computation technique with regard to its accuracy. To explain this, one must consider the ability of ANN algorithms to model complex relationships [35] between input and output variables. In addition, ANN can handle imprecise and fuzzy data, whether ordinary, categorical, or binary, without violating any assumptions [50,51], while LR is limited in its ability to capture non-linear relationships between the explained and explanatory variables [52]. Therefore, because the rockfall propagation zone confuses the logit model and hinders its ability to estimate the intercept and correlation parameters of the model effectively, the performance of the latter is negatively affected. On the other hand, the multilayer, multistep ANN algorithm introduced some changes and differences to the models without altering its accuracy. This is due to its ability to generalize from the whole data set rather than focusing on specific data points [53]. Conversely, LR is sensitive to outliers and noisy data. This sensitivity can lead to inaccurate and inconsistent results in modeling rockfall susceptibility, which are more significant compared to other landslide types [54].

## 7. Conclusions

Rockfall mapping is a difficult and laborious task that requires good knowledge of morphological features. This type of landslide is particularly difficult to model because the source and depositional areas are not similar concerning their geological and geomorphological characteristics. The findings of this paper constitute preliminary results that suggest significant variability in the output of the models when different input areas used. Therefore, the preparation of input data for rockfall susceptibility modeling must be based on the desired outcome. If the researcher is more interested in the initiation of rockfalls, he/she must only use the source area to build the RS model. Otherwise, the assessment of the weight of rockfall conditioning factors may lead to inaccurate or erroneous values. However, if one is attempting to model RS for hazard prevention and management purposes, source area-derived models may produce very conservative models that may not be very useful, especially given the fact that the rockfall deposition area is the inhabited portion of this kind of landslide.

**Author Contributions:** Conceptualization, Y.E.M., Y.E.K., R.E.H. and A.B.; methodology, Y.E.M. and A.B.; software, Y.E.M.; validation, Y.E.M., A.B. and Y.E.K.; formal analysis, Y.E.M. and A.B.; investigation, Y.E.M.; resources, Y.E.K. and R.E.H.; writing—preparation of the original version, Y.E.M.; writing—reading and editing, Y.E.M., Y.E.K., A.B. and R.E.H.; supervision, Y.E.K. and R.E.H.; project administration, Y.E.K.; funding acquisition, Y.E.K. and R.E.H. All authors have read and agreed to the published version of the manuscript.

**Funding:** The authors of this paper wish to express their sincere appreciation for the financial support received from CNRST within the framework of the research project PPR2/205/65. We would also like to thank the journal's editors and reviewers for kindly considering and revising this manuscript.

**Institutional Review Board Statement:** Not applicable.

**Informed Consent Statement:** Not applicable.

**Data Availability Statement:** The raw data supporting the conclusions of this article will be made available by the authors on request.

**Conflicts of Interest:** The authors declare no conflicts of interest.

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
