# Peer review of "Effect of Rockfall Spatial Representation on the Accuracy and Reliability of Susceptibility Models (The Case of the Haouz Dorsale Calcaire, Morocco)"

_land, doi:10.3390/land13020176_

Round 1

Reviewer 1 Report

Comments and Suggestions for Authors

The susceptibility analysis of rockfalls in northern Morocco is presented in the paper. The authors used logistic regression and multi-layer perceptron to assess the endanger of slopes considering two different kinds of input data sets. I have following remarks:

1. It is not clear what proportion of rockfall and non-rockfall areas was (balanced or imbalanced)? 

2. The authors should use different types of metrics to evaluate the quality of used methods (e.g. precision, accuracy, recall, f1_score).

3. I suggest to use Shapley value to explain which input parameters have the significant impact in the Multi-layer Perceptron model.

4. Fig. 4 - I do not see where the testing area is.

5. The Conclusion chapter should be expanded. Sentences summarizing the Authors' research are too general and do not provide new knowledge.

Author Response

Response to Reviewer 1

The susceptibility analysis of rockfalls in northern Morocco is presented in the paper. The authors used logistic regression and multi-layer perceptron to assess the endanger of slopes considering two different kinds of input data sets. I have following remarks:

1. It is not clear what proportion of rockfall and non-rockfall areas was (balanced or imbalanced)? 

The data is imbalanced. The non-rockfall area is naturally larger than the rockfall-affected area since rockfalls are rare in nature compared to other landforms.

2. The authors should use different types of metrics to evaluate the quality of used methods (e.g. precision, accuracy, recall, f1_score).

Thank you for this suggestion. However, the aim of the paper is not to evaluate the quality of the output model. We mainly attempt to find out what type of variability is observed with two different algorithms. The use of f1_score is not possible given the fact that such techniques are meant to evaluate the accuracy of deterministic models not that of probabilistic ones. 

3. I suggest to use Shapley value to explain which input parameters have the significant impact in the Multi-layer Perceptron model.

Thank you for this suggestion. The synoptic weight of each variable was used according to your suggestion (please check the revised version).

4. Fig. 4 - I do not see where the testing area is.

We have improved the visibility of the testing dataset in the Figure 4 (please check the revised version).

5. The Conclusion chapter should be expanded. Sentences summarizing the Authors’ research are too general and do not provide new knowledge.

We have rewritten the conclusion section according to your suggestions.

(please check the revised version).

Reviewer 2 Report

Comments and Suggestions for Authors

In Introduction, articles related to the study and the study area is Rif mountains (Morocco), should be considered.

Figure 2 must be corrected. Both testing and training data proportions are 70%. Testing dataset must be 30%.

I do not understand the physical meaning of distance to stream with rockfall. This parameter is a little bit confusing.

Rainfall is not a rockfall condition factor. It must be a trigger. Rainfall must be eliminated from the conditioning factors.

Rockfall susceptibility is not understandable. “Rockfall source area susceptibility” and “rockfall accumulation area susceptibility” should be used.

Conclusion is highly poor. This section must be enhanced employing the scientific highlights obtained from the study. 

Comments on the Quality of English Language

Proofreading is necessary.

Author Response

Response to Reviewer 2

1. In Introduction, articles related to the study and the study area is Rif mountains (Morocco), should be considered.

We have added some references to the introduction that deal with susceptibility analysis and, that are related to our study area (the Rif mountains). These references can further enrich this section of work:

Milliès-Lacroix, “L’instabilité des versants dans le domaine rifain,” Revue Géomorphologie Dynamique, vol. 15, no. 7-8–9, pp. 97–109, 1995.

Fonseca, “Large deep-seated landslides in the northern Rif Mountains (Northern Morocco) : inventory and analysis,” 2014.

C. Ozer et al., “On the use of hierarchical fuzzy inference systems (HFIS) in expert-based landslide susceptibility mapping: the central part of the Rif Mountains (Morocco),” Bulletin of Engineering Geology and the Environment, vol. 79, no. 1, pp. 551–568, Jan. 2020, doi: 10.1007/s10064-019-01548-5.

Obda, A. Bounab, K. Agharroud, R. Sahrane, and Y. El Kharim, “A multidisciplinary approach to investigate active and new tectonic effects on landslides spatial distributions: case study in the Pre-Rif Ridges morphostructural unit,” Natural Hazards, 2023, doi: 10.1007/s11069-023-06243-z.

Prokos, H. Baba, D. Lóczy, and Y. El Kharim, “Geomorphological hazards in a Mediterranean mountain environment - Example of tétouan, Morocco,” Hungarian Geographical Bulletin, vol. 65, no. 3, pp. 283–295, 2016, doi: 10.15201/hungeobull.65.3.6.

EL Kharim, “Geological features of the slope instability in Tetouan region (the northern Rif, Morocco),” Bol. R. Soc. Esp. Hist. Nat. Sec. Geol, vol. 106, pp. 39–52, 2012.

Marouane Benmakhlouf, Y. El Kharim, J. Galindo-Zaldivar, and R. Sahrane, “Landslide Susceptibility Assessment in Western External Rif Chain using Machine Learning Methods,” Civil Engineering Journal, vol. 9, no. N 12, 2023, doi: http://dx.doi.org/10.28991/CEJ-2023-09-12-01.

Harmouzi, H. A. Nefeslioglu, M. Rouai, E. A. Sezer, A. Dekayir, and C. Gokceoglu, “Landslide susceptibility mapping of the Mediterranean coastal zone of Morocco between Oued Laou and El Jebha using artificial neural networks (ANN),” Arabian Journal of Geosciences, vol. 12, no. 22, Nov. 2019, doi: 10.1007/s12517-019-4892-0.

Bounab, Y. El Kharim, R. El Hamdouni, and R. Hlila, “A multidisciplinary approach to study slope instability in the Alboran Sea shoreline: Study of the Tamegaret deep-seated slow-moving landslide in Northern Morocco,” Journal of African Earth Sciences, vol. 184, p. 104345, 2021, doi: https://doi.org/10.1016/j.jafrearsci.2021.104345.

Obda, Y. El Kharim, I. Obda, and A. El Kou, “Coastal rocky slopes instability analysis and landslide frequency-area distribution alongside the road network in west Mediterranean context (Northern of Morocco),” 2022, doi: 10.21203/rs.3.rs-2196461/v1.

2. Figure 2 must be corrected. Both testing and training data proportions are 70%. Testing dataset must be 30%.

We have made the necessary corrections to the proportion of test and training data in the figure following your request. Specifically, we have adjusted the distribution to 30% and 70% for the test and training data sets, respectively, as shown in the new figure in the document.

3. I do not understand the physical meaning of distance to stream with rockfall. This parameter is a little bit confusing.

Water streams can significantly increase the steepness of natural slopes especially in Rocky terrain. Therefore, distance to the water stream is an indirect measure of the influence of water erosion on the output RSMs.

4. Rainfall is not a rockfall condition factor. It must be a trigger. Rainfall must be eliminated from the conditioning factors.

As far as the rainfall variable is concerned, it's true that rainfall is a trigger for landslides in general, but the use of rainfall is very important in a landslide susceptibility analysis[1], particularly in mountainous areas. In susceptibility analysis, only the spatial component of mean annual precipitation and its variations across the study area are used. ie. [2-3].

Normally, it is the intensity of rainfall in 24 hours that is used as a triggering factor while the spatial variability of mean annual rainfall can be used as determining factor for slope movements in an area.

[1] M. E. A. Budimir, P. M. Atkinson, and H. G. Lewis, “A systematic review of landslide probability mapping using logistic regression,” Landslides, vol. 12, no. 3. Springer Verlag, pp. 419–436, Jun. 01, 2015. doi: 10.1007/s10346-014-0550-5.

[2] M. Arab Amiri and C. Conoscenti, “Landslide susceptibility mapping using precipitation data, Mazandaran Province, north of Iran,” Natural Hazards, vol. 89, no. 1, pp. 255–273, 2017, doi: 10.1007/s11069-017-2962-8.

[3] Q. Lin et al., “National-scale data-driven rainfall induced landslide susceptibility mapping for China by accounting for incomplete landslide data,” Geoscience Frontiers, vol. 12, no. 6, p. 101248, 2021, doi: https://doi.org/10.1016/j.gsf.2021.101248.

5. Rockfall susceptibility is not understandable. “Rockfall source area susceptibility” and “rockfall accumulation area susceptibility” should be used.

The accumulation zone is affected by multiple displacement factors, such as runoff, which can carry materials over long distances and deposit them in specific areas such as channels or the base of embankments. This zone is an essential tool for assessing the potential risk of damage not only to property but also to human life. The main distinction between the propagation zone and the accumulation zone is that the former is where boulders can break loose, while the latter is where the accumulated mass of boulders can pose a potential threat.

6. Conclusion is highly poor. This section must be enhanced employing the scientific highlights obtained from the study. 

the article's conclusion is rewritten in the revised version, highlighting the study's main scientific findings.

7. Comments on the Quality of English Language: Proofreading is necessary.

The manuscript has been proofread, with several corrections to improve its comprehensibility. 

We have also made minor changes to the language and article structure to improve clarity and fluidity.

Round 2

Reviewer 2 Report

Comments and Suggestions for Authors

The revisions are satisfactory and hence it can be accepted without further revision.

Comments on the Quality of English Language

The revisions are enough and hence it can be accepted without further revision.